# Skilful predictions of the Asian summer monsoon one year ahead

Yuhei Takaya [1✉], Yu Kosaka [2], Masahiro Watanabe [3] & Shuhei Maeda[4]

The interannual variability of the Asian summer monsoon has significant impacts on Asian society. Advances in climate modelling have enabled us to make useful predictions of the seasonal Asian summer monsoon up to approximately half a year ahead, but long-range predictions remain challenging. Here, using a 52-member large ensemble hindcast experiment spanning 1980–2016, we show that a state-of-the-art climate model can predict the Asian summer monsoon and associated summer tropical cyclone activity more than one year ahead. The key to this long-range prediction is successfully simulating El Niño-Southern Oscillation evolution and realistically representing the subsequent atmosphere–ocean response in the Indian Ocean–western North Pacific in the second boreal summer of the prediction. A large ensemble size is also important for achieving a useful prediction skill, with a margin for further improvement by an even larger ensemble.

[1] Meteorological Research Institute, Japan Meteorological Agency, Ibaraki, Japan. [2] Research Center for Advanced Science and Technology, The University of Tokyo, Tokyo, Japan. [3] Atmosphere and Ocean Research Institute, The University of Tokyo, Chiba, Japan. [4] Aerological Observatory, Japan Meteorological Agency, Ibaraki, Japan. ✉email: yuhei.takaya@mri-jma.go.jp

The variability of the Asian summer monsoon has considerable impacts on the human lives and economy throughout Asia[1], the most populous region on the globe, by modulating seasonal precipitation, surface temperatures, and the occurrences of floods[2], droughts[3] and tropical cyclones (TCs)[4]. Summer rainfall and the discharge of major rivers in the Asian monsoon region are vital for water security[5] and food production[1,6]. Accurately predicting the Asian summer monsoon with a long lead time is thus of great value for decision-making across a wide range of sectors[1–8].

Despite considerable amounts of effort, long-range Asian summer monsoon predictions have presented immense challenges[9–13]. Complex atmosphere–land–ocean interactions, global interbasin interactions and unpredictable atmospheric internal variabilities limit model representations and predictions of the Asian monsoon[9–11,14–16]. In particular, anomalous precipitation over the tropical western North Pacific (WNP) and South China Sea, an essential component of Indo-WNP summer monsoon variability, is negatively correlated with local sea surface temperature (SST) in summer (Supplementary Fig. 1), indicating dominance of local atmospheric forcing on the ocean and suggesting lack of potential predictability from local SST[17,18].

However, improvements to climate models in representing atmosphere–ocean–land processes and initialisation techniques have steadily extended seasonal predictions and beyond[8,19,20], and certain capabilities have been achieved in predicting the Asian monsoon with up to about half a year in advance[11–13,19–22]. Moreover, recent advances in understanding the mechanism of the Indo-WNP summer monsoon variability suggest seasonal predictions with longer leads than previously expected[23–26].

Here, we demonstrate the capability of a long-range prediction of the Indo-WNP summer monsoon far beyond current seasonal predictions. We also illustrate that the skilful prediction of the Asian summer monsoon 1 year ahead indeed stems from successful simulations of the El Niño-Southern Oscillation (ENSO) evolution and the ENSO-induced subsequent atmosphere–ocean variation in the Indian Ocean–WNP.

## Results and discussion

**Prediction skill of the Asian summer monsoon 1 year ahead.** We conducted a 16-month-long prediction experiment using the quasi-operational seasonal prediction system called the Japan Meteorological Agency/Meteorological Research Institute-Climate Prediction System version 2 (JMA/MRI-CPS2)[21]. A set of 52-member ensemble hindcasts starting from every April was established for the summer seasons for 37 years from 1980 to 2016. We particularly focused on the prediction skill for the second-year boreal summer (June–August with a 13-month lead) against historical observations and reanalysis (Methods).

JMA/MRI-CPS2 skilfully predicts key indices representing the interannual variability of the Indo-WNP monsoon and ENSO[27] 1 year ahead (Fig. 1; see Methods for the index definitions). Ensemble envelopes (maximum–minimum ranges) generally encompass the observations by virtue of the large ensemble size. The WNP monsoon index, which represents the dominant variability of the WNP monsoon based on 850 hPa zonal wind[28], is predicted at a significant correlation skill of $r = 0.50$ ($p < 0.005$) for the second summer (Fig. 1a; see Methods for the skill evaluation). The predictability of the Indo-WNP monsoon originates from a slowly evolving SST[8] but not necessarily through a direct influence from concurrent ENSO conditions. Indeed, the prediction skill of the NINO3.4 SST index for the second summer is only moderate ($r = 0.41$, $p = 0.012$; Fig. 1b)

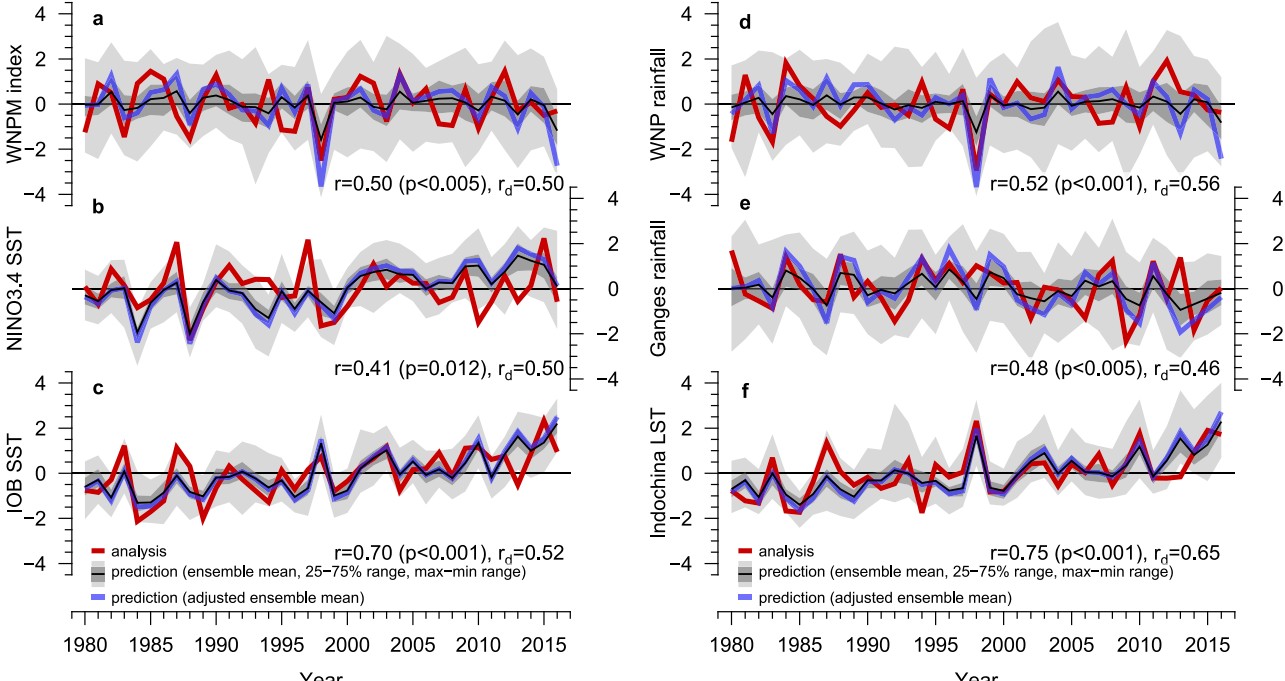

**Fig. 1 Prediction skill for climate indices in the second summer.** Time series of **a** the western North Pacific monsoon (WNPM) index, **b** NINO3.4 sea surface temperature (SST) index, **c** Indian Ocean Basin (IOB) SST index, **d** western North Pacific (WNP) rainfall index, **e** Ganges rainfall index, and **f** Indochina land surface (2 m air) temperature (LST) index (see Methods for definitions). JMA/MRI-CPS2 ensemble mean predictions (thin black lines) and observations (thick red lines) are presented with a maximum–minimum and interquartile ranges of the ensemble predictions (grey shading). The ensemble mean predictions and observations are normalised by the mean and standard deviation of the concatenated all-member predictions and observations. Adjusted ensemble mean predictions (thick blue lines) are those normalised by the mean and standard deviation of the ensemble mean predictions. Correlation coefficients between the ensemble mean predictions and observations are shown at the bottom right ($r$ raw time series, $r_d$ after linear detrending).

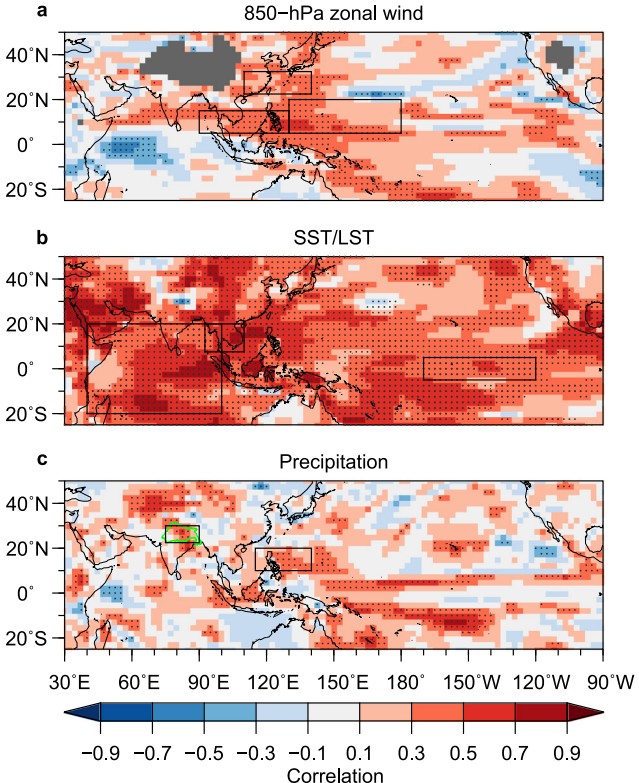

**Fig. 2 Prediction skill for the second summer Asian monsoon.** Correlation coefficients between the observations and ensemble mean predictions for **a** 850 hPa zonal wind, **b** sea surface temperature (SST) over the ocean and land surface (2 m air) temperature over land (LST), and **c** precipitation. Stippled regions are statistically significant at the 5% level according to Student's $t$ test. Boxes indicate the regions of indices used in this study (Methods). In (**c**), the Ganges basin[53] is shown in green.

due to the so-called spring predictability barrier[29]. Instead, the Indian Ocean Basin (IOB)-wide SST variability, which is predicted with a much higher correlation skill of $r = 0.70$ ($p < 0.001$) owing to a delayed Indian Ocean response to preceding ENSO (Fig. 1c)[23,26], is fundamental to predicting the Indo-WNP summer monsoon (discussed later). The correlation skill for the IOB SST remains significant even after linear detrending ($r_d = 0.52$). Notably, the correlation skill for the IOB SST drops slightly for the first few months but recovers and remains high until the second summer (Supplementary Fig. 2). It is also noted that the model presents higher prediction skill of NINO3.4 SST with linear detrending than without, due to an overestimate of the warming trend of NINO3.4 SST in the model (Fig. 1b), consistent with a multi-model ensemble coupled seasonal predictions[30].

The WNP monsoon index is significantly correlated with surface temperature and precipitation over broad regions of Asia[23,24,26] (Supplementary Fig. 3), highlighting the importance of its skilful prediction[23,24]. With a statistically significant skill for predicting the WNP monsoon circulation for the second summer, the model predicts tropical WNP rainfall with a high statistical significance ($r = 0.52$, $p < 0.001$; Figs. 1d and 2c), despite the lack of a local SST predictability source (Supplementary Fig. 1). There is a markedly high skill ($r = 0.75$, $p < 0.001$; Fig. 1f) for the 2 m land surface air temperature over Indochina (mainland Southeast Asia), reflecting a strong influence from the Indo-Pacific Ocean. In addition, we find meaningful prediction skills for precipitation in the Ganges River Basin ($r = 0.48$, $p < 0.005$; Fig. 1e) as well as around Indonesia and the Horn of Africa (Fig. 2c). In particular, the reliable prediction of precipitation in the Ganges river basin is

of primary importance and has considerable implications for water resources management[5]. Previous studies suggested that precipitation in the Ganges river basin is associated with Asian monsoon variability[31,32].

Pointwise temporal correlation maps further corroborate the prediction skills (Fig. 2). Notably, distributions of relatively high correlations for 850 hPa zonal wind and precipitation match the dominant pattern of variability associated with the Indo-WNP summer monsoon[23,26,28] as represented here by the WNP monsoon index (Supplementary Fig. 3). These correlations are also generally consistent with the inherent potential predictability (Supplementary Fig. 4). Notably, the model retains meaningful prediction skill after linear detrending, indicating its capability to predict the interannual variability (Supplementary Fig. 5). Moreover, the contribution of the trend to the actual prediction skill is also consistent with its contribution to the potential predictability (Fig. 2; Supplementary Figs. 4 and 5). We additionally note that the model predicts the first summer with generally higher skill (Supplementary Fig. 6).

**Underlying mechanisms of the skilful predictions.** Having obtained skilful 1-year-lead predictions of the Asian summer monsoon, we then discuss the mechanism by which the Indian Ocean mediates the El Niño influence on the Asian climate in the subsequent summer[23,26] (Supplementary Fig. 7). Similar to a battery charging a capacitor, El Niño warms the Indian Ocean from its peak boreal winter to spring through changes in the Walker circulation and Indonesian throughflow. While El Niño SST anomalies in the equatorial eastern Pacific typically disappear by the subsequent boreal summer, Indian Ocean warm conditions persist and, like a discharging capacitor, trigger coherent ocean–atmosphere variability called the Indo-western Pacific Ocean capacitor (IPOC) mode. In the IPOC mode, the warmer Indian Ocean excites an atmospheric Kelvin wave response and induces surface Ekman divergence over the tropical WNP, where atmospheric convections are suppressed. In response, an anomalous lower-tropospheric anticyclone corresponding to a weak WNP summer monsoon extends westward and affects the Indo-WNP climate while providing feedback to SST warming in the North Indian Ocean and the tropical WNP west of 150°E. SST cooling in the tropical WNP east of 150°E also amplifies these anomalies through wind–evaporation–SST feedback[24,25,33]. ENSO and the subsequent IPOC development constitute a year-long process, and the latter has pervasive influences on the Asian climate and seasonal TC activity[23,24,31,34,35]. Therefore, ENSO growth in the first year and subsequent IPOC development is likely the key to the year-long predictability of the Asian monsoon[19,23,26].

The above hypothesis is confirmed by a simple skill evaluation: the prediction performance of IOB SST and associated WNP summer monsoon tends to decrease when summers following major El Niños are excluded (Supplementary Table 1). By contrast, such a decrease does not occur when we exclude the summers of developing major El Niños. These findings indicate that preceding El Niños contribute more to the prediction skill for the second summer than concurrent El Niños.

Composite maps for boreal summers after major El Niños (observed NINO3.4 index >1 std. dev. in preceding November–January; 1983, 1992, 1998, 2003, 2010 and 2016; Fig. 3) further substantiate that the ENSO–IPOC coupling conveys successful second summer predictions for the Indo-WNP and Asian climate. Those summers exhibit high surface pressure and suppressed rainfall over the tropical WNP and enhanced rainfall around the Maritime Continent, consistent

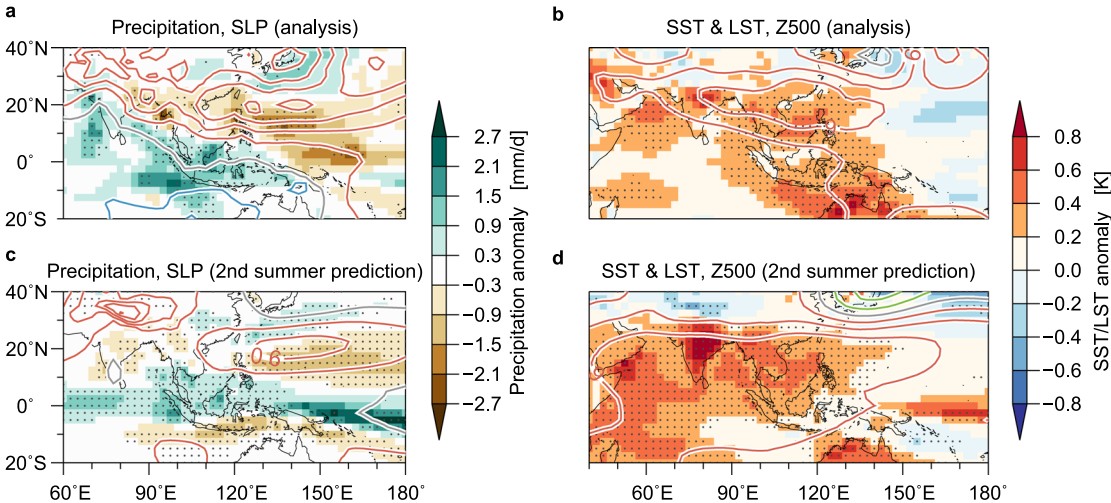

**Fig. 3 The IPOC mode and its second summer prediction.** Composite anomalies of the (**a**, **b**) observations and (**c**, **d**) JMA/MRI-CPS2 13-month-lead prediction for summers following major El Niño events (1983, 1992, 1998, 2003, 2010 and 2016; see text for definition). **a**, **c** Precipitation (colours) and sea level pressure (SLP; contours with an interval of 0.3 hPa; red for positive, grey for zero and blue for negative). **b**, **d** SST over the ocean and land surface (2 m air) temperature (LST) over land (colours) and 500 hPa geopotential height (Z500; contours with an interval of 3 m; red for positive, grey for zero and green for negative). Stippled regions are statistically significant at the 5% level based on a bootstrap method (10,000 resamplings).

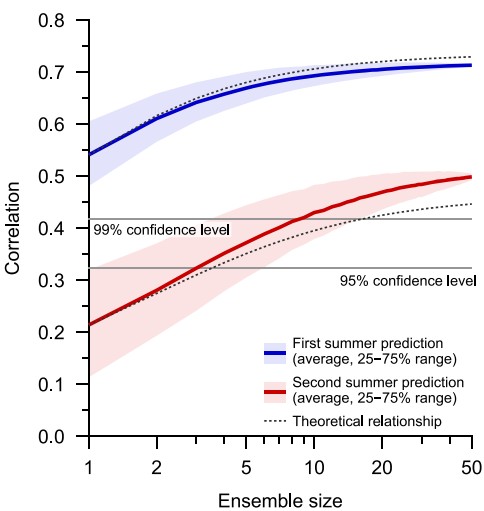

**Fig. 4 Prediction skill dependency on ensemble size.** Correlation coefficients of the WNP monsoon index for the first (blue line) and second (red line) summers. Medians (thick lines) and 25–75% ranges (colour shading) are estimated by using a bootstrap method (10,000 resamplings of ensemble members with a fixed retrospective prediction period). The theoretical estimates[45] are shown based on averages of 1-member correlation skills (Methods). The statistical significance levels are based on Student's *t* test.

with previous studies[23,26]. Warmer surface temperatures are remarkable in India and Southeast Asia, surrounded by warmer SSTs. In addition, the 500 hPa height anomalies over the WNP feature a poleward teleconnection called the Pacific–Japan pattern[36], increasing predictability over midlatitude East Asia[23,26] (Supplementary Fig. 3). We find that these observed IPOC-related teleconnections are reasonably well reproduced in the second summer predictions (Fig. 3 and Supplementary Fig. 3). Further-more, the WNP summer monsoon is enhanced by concurrent El Niños[24,32] and suppressed by La Niñas, indicating additional contributions from accurate ENSO predictions through the second summer to successful Indo-WNP monsoon predictions. The model, however, presents a tendency of a weaker transition

from El Niño to La Niña than observed, which may limit the prediction skill of the WNP summer monsoon (Fig. 3). The lag composites of the observations and predictions after the major El Niños further support the model capability of reproducing and predicting the IPOC development along the lifecycle of ENSO (Supplementary Fig. 7). In summary, the success of the 1-year-lead prediction stems from the model's ability to predict ENSO evolution and IPOC development in the second summer as well as its capability to reproduce the associated climate anomalies.

**Ensemble size dependency of the prediction skill.** Apart from the predictability inherent to the atmosphere–ocean system, large ensembles are generally necessary for obtaining meaningful prediction skills for phenomena featuring low signal-to-noise ratios[37,38]. This holds true for the 1-year-lead prediction of the Asian summer monsoon. Figure 4 presents the ensemble size dependency of the prediction skill. The correlation coeffi-cients for the first summer prediction are well above the 99% confidence level, indicating the high fidelity of the seasonal pre-diction. For the second summer prediction, by contrast, the correlation coefficients are far below the 99% confidence level when the ensemble size is small (<10 members), while a statis-tically significant correlation skill is achievable with a large ensemble size (>20 members). This skill increase reasonably agrees with a theoretical estimate (Methods). The second summer performance does not fully level off around the maximum ensemble size of the present study, suggesting possible further enhancement of the skill by increasing the ensemble size, albeit with a slower rate. We also note that the estimated potential predictability (Supplementary Fig. 5) is overall higher than the real prediction skills. Thus, the so-called signal-to-noise paradox[38] is not apparent in the 1-year-lead Asian summer monsoon prediction in JMA/MRI-CPS2.

**WNP TC predictions.** Finally, we discuss the capability of sea-sonal TC predictions in the WNP (Methods). The model can make predictions of the WNP TC density averaged over the WNP (0°–60°N,100°E–180°) during June–August, which corresponds to the first half of the WNP TC season[34], with a highly significant

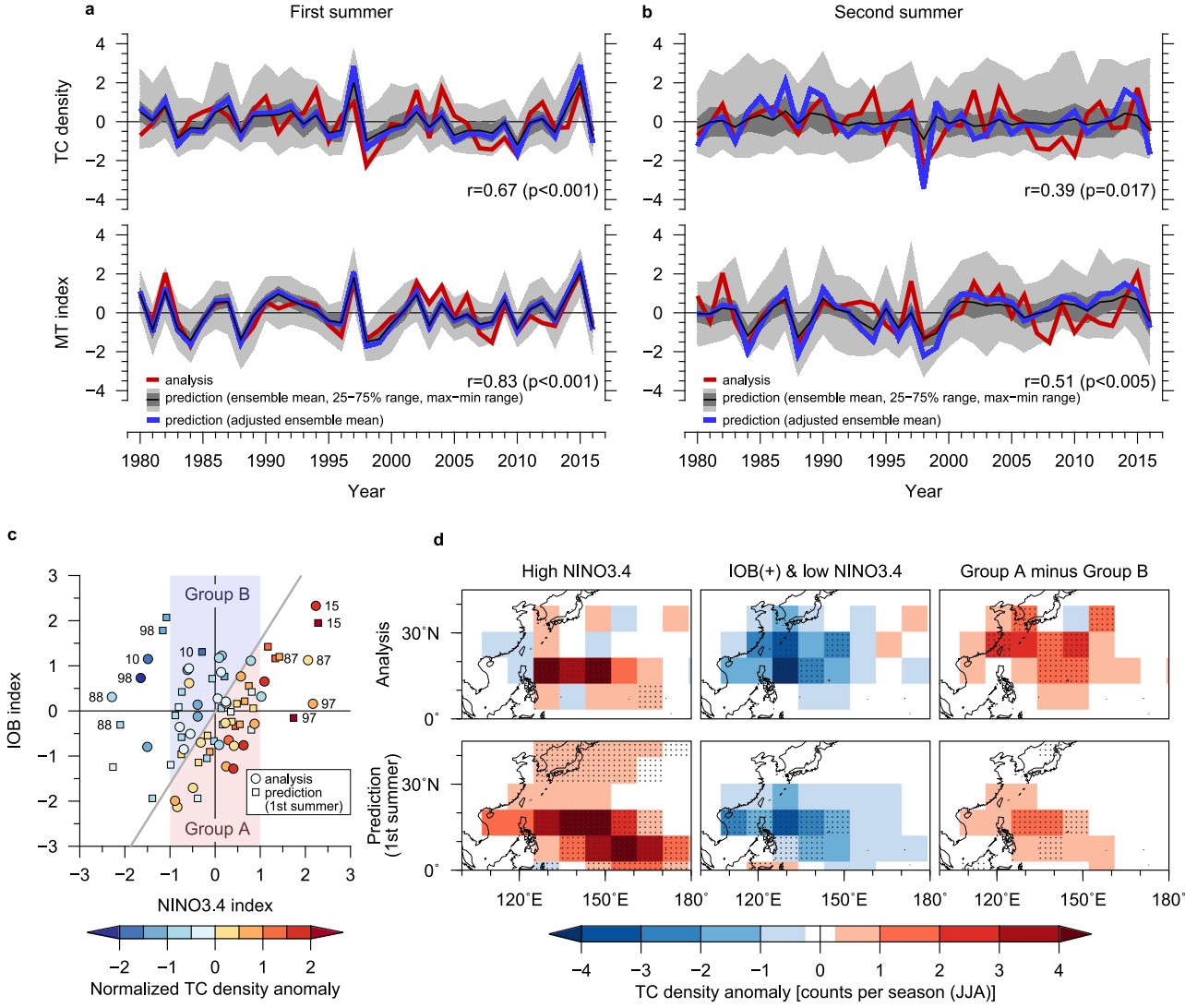

**Fig. 5 TC prediction skill and influence from ENSO and IOB SST.** Same as Fig. 1 but for the **a** first and **b** second summer predictions of the tropical cyclone (TC) density and monsoon trough (MT) index. **c** Scatterplot of the analysed (circles) and predicted (squares) TC densities with respect to their own Indian Ocean Basin (IOB) and NINO3.4 SST indices. Colours indicate the TC density accumulated in the western North Pacific (WNP) region normalised by the climatological mean and standard deviation. Two-digit numbers shown alongside circles or squares indicate years. **d** Composites of observed and predicted TC density anomalies for summers with (left) the NINO3.4 index >+1.5 std. dev., (middle) the IOB index >0 and NINO3.4 index <−1 std. dev., and (right) Group A minus Group B (groups shown in Fig. 5c). Stippled regions are statistically significant at the 5% level according to a bootstrap method (10,000 resamplings).

skill ($r = 0.67$, $p < 0.001$) for the first summer (Fig. 5a) and a moderate skill ($r = 0.39$, $p = 0.017$) for the second summer (Fig. 5b). We note that the skill is slightly lower for July–September in the second year ($r = 0.35$, $p = 0.035$) than for June–August, due to gradual dissipation of delayed ENSO influence and skill decline with a lead time[23,35].

The seasonal WNP TC activity (TC density) is predominantly modulated by the WNP monsoon trough (MT) strength (Methods)[39,40]. The model effectively captures the observed variability of the MT index ($r = 0.83$, $p < 0.001$ for the first summer; $r = 0.51$, $p < 0.005$ for the second summer; Figs. 5a, b). This MT prediction skill underscores the model's capability of predicting WNP TC activity[40]. The prediction captures the suppressed MT strength and TC activity in 1988 and 1998, which are marked IPOC years following strong El Niños (Fig. 3b). The ensemble envelopes generally encompass the observed TC activity and MT index. The second summer prediction failure for 2010 is presumably due to a failure to predict the 2009/2010 El Niño (Fig. 1b).

The IPOC and ENSO exhibit competing effects on summer WNP TC activity through MT changes. On one hand, warmer Indian Ocean conditions suppress TC activity by weakening the WNP MT as part of the IPOC[23,34,35]. On the other hand, the warm ENSO phase shifts the MT (and thereby the TC genesis locations) southeastward, leading to a longer TC lifetime[41,42]. Figure 5c confirms these IPOC and ENSO influences in the observations and the first summer predictions. Notably, these effects are present even in moderate ENSO cases (with the NINO3.4 index within ±1 std. dev.); in Fig. 5c, the two groups (differentiated by background colours) are classified by signs of a predictand (TC density) of a bivariate linear regression model based on observational data (Methods). The TC density anomalies (colours) of the predictions are in good agreement with the observations, supporting that both ENSO and the IOB SST play roles in modulating TC activity (Fig. 5c).

Our composite analysis (Fig. 5d; Methods) elaborates the observed relationship. In strong El Niño summers, TC activity is enhanced, particularly in the southeastern WNP (Fig. 5d, left). In

La Niña summers with warm IOB SSTs, TC activity is suppressed broadly throughout the WNP (Fig. 5d, middle). The distinct effects of the IOB SST and ENSO on TC activity are also confirmed even for cases of neutral to weak ENSO summers in the composite difference between Groups A and B (Fig. 5d, right) classified based on the bivariate linear regression (Fig. 5c). A correlation matrix further confirms the competing effects of the IOB SST and ENSO (Supplementary Fig. 8). In summary, the successful prediction of summer TC activity arises from the extended predictability of the WNP MT associated with the IPOC mode and ENSO. In the second summer prediction, these processes are reproduced in some marked ENSO and IPOC years, giving a moderate but meaningful prediction skill for TC activity (Fig. 5b).

In summary, state-of-the-art climate modelling has the potential to overcome the difficulty in predicting the Asian summer monsoon. Based on a large ensemble hindcast experiment, we have demonstrated the capability to make skilful Indo-WNP monsoon predictions with a lead time exceeding 1 year. Precipitation, surface temperature and circulation associated with the Indo-WNP summer monsoon and WNP TC activity in summer can be predicted with meaningful skill, consistent with the estimated potential predictability inherent to the climate system.

The delayed ENSO influence mediated by Indian Ocean anomalies (the IPOC mode) is the primary mechanism for establishing a skilful 1-year-lead prediction. The model veracity at reproducing IPOC-related climate anomalies is also fundamental for skilful predictions. Feasible TC predictions benefit additionally from the remote influence of the central–eastern equatorial Pacific SST associated with concurrent ENSO conditions. The prediction skill may be further improved not only by increasing the ensemble size but also by using models with higher capabilities of predicting ENSO and its delayed impacts in the WNP, i.e., the IPOC mode. Our results present promise for further long lead seasonal predictions in Asia with bright prospects for extensive applications.

## Methods

**Prediction experiment.** A 16-month-long ensemble prediction experiment with 52 members starting from April spanning 38 years (1979–2016) is conducted using the JMA/MRI-CPS2 seasonal prediction system based on an atmosphere–ocean–sea ice–land coupled model[21]. The 52-member ensemble consists of a series of 13-member predictions initialised on four calendar dates (April 11, 16, 21 and 26). The model has an atmospheric resolution of ~110 km in the mid-latitudes, 60 atmospheric levels with its top at 0.01 hPa, and an oceanic resolution of 0.5° × 1.0° with latitudinal grid refinement near the equator (0.3°). The ocean and atmospheric initial conditions are produced by an ocean analysis system (MOVE/MRI.COM-G2)[43] and atmospheric analysis system (JRA-55)[44]. The land initial conditions are taken from the JRA-55 land analysis. The ensemble members are generated by using slightly perturbed initial conditions and employing a stochastic physics scheme[45]. More details can be found in Takaya et al[21].

**Observational data.** To evaluate and analyse the predictions, we use JRA-55 reanalysis data[44] for winds, geopotential height, 2 m air temperature and sea level pressure, COBE-SST data[46] for SST, and GPCP version 2.3 data[47] for precipitation. All data are interpolated on a regular 2.5° grid for the analyses except for the TC analysis, which requires a finer resolution, and thus, the data are regridded to a 1.5° resolution. We additionally use TC analysis data (best track data) provided from RSMC Tokyo to examine the interannual variability of seasonal TC activity. TCs stronger than tropical storms (maximum winds exceeding 17.2 m s$^{-1}$) are analysed and compared with the predictions in this study.

**Climate indices used for analysis.** Several indices representing the dominant interannual variability of SST, atmospheric circulations and TC activity are computed for this study. The NINO3.4 index is defined as SST anomalies averaged over the central to eastern Pacific (5°N–5°S, 170°W–120°W) representing ENSO variability[27], and the IOB index is defined as SST anomalies averaged over the Indian Ocean (20°N–20°S, 40°E–100°E)[26]. The WNP summer monsoon circulation index (WNP monsoon index) is defined as the 850 hPa zonal wind difference

between the northern (22.5°N–32.5°N, 110°E–140°E) and southern (5°N–15°N, 90°E–130°E) regions[48]. Despite its naming, the southern box of the WNP monsoon index extends to the eastern Indian Ocean, and the index thus captures the summer monsoon variability throughout the Indo-WNP (Supplementary Fig. 3). The Ganges and WNP rainfall indices are defined as rainfall averaged over 22.5°N–30°N, 75°E–90°E and 10°N–20°N, 115°E–140°E, respectively[48]. The Indochina temperature is defined as the 2 m land surface air temperature averaged over 7.5°N–20°N, 92.5°E–110°E. All indices are normalised with climatological means and standard deviations.

**Prediction skill evaluation.** The Pearson correlation coefficient (denoted as $r$) between the observations and ensemble mean prediction is used to evaluate the prediction skill. The $p$ value (denoted as $p$) is determined by referring to Student's $t$ distribution.

**Theoretical estimation of the skill dependence on ensemble size.** The equation for estimating the theoretical skill (correlation coefficient) as a function of the ensemble size is derived under the perfect model assumption[49] as

$$C_{M} = \frac{M^{1/2}C_1}{[1 + (M-1)C_1]^{1/2}}, \tag{1}$$

where $M$ is a given ensemble size, $C_1$ is an expectation of a correlation coefficient skill of a single-member prediction, and $C_M$ is an expectation of an $M$-member ensemble mean prediction.

**Evaluation of TC activity.** To evaluate the seasonal TC activity for June–August, we calculate the TC density using 6-hourly data. TCs with maximum wind exceeding 17.2 m s$^{-1}$ are examined here. Model TCs are detected using an objective detecting and tracking method[35,50], which is similar to the methods applied in previous studies[51]. The objective detection method is applied to 6-hourly model outputs at a 1.5° × 1.5° resolution with the following conditions and criteria.

1. A grid point with a local sea level pressure minimum in a 6° × 6° box over the ocean between the equator and 30°N is determined as the centre of a candidate TC.
2. The relative vorticity at 850 hPa is below $5 \times 10^{-5}$ s$^{-1}$ in a 3° × 3° box surrounding the centre of a TC.
3. The geopotential height thickness between 200 and 500 hPa at the centre of the candidate TC is 7 gpm higher than the average thickness in a 9° × 9° box surrounding the centre of the TC, excluding the centre of the candidate TC (24 grid points).
4. At the centre of the candidate TC, the wind speed at 200 hPa is lower than that at 850 hPa.

The four conditions above must hold continuously for at least 12 h for a TC to be detected. These thresholds have been chosen so that the number of detected TCs matches the observations of the RSMC Tokyo best track analysis with maximum winds exceeding 17.2 m s$^{-1}$. For Fig. 5d, the TC density is defined as the 6-hourly TC count in each 4.5° × 4.5° box.

For tracking, TCs are searched in 9° × 9° boxes around previous TC positions. Two days after TC genesis, only criteria 3 and 4 are applied, and the relative vorticity threshold is reduced to $3.5 \times 10^{-5}$ s$^{-1}$.

A previous study validated the above algorithm using JRA-25 reanalysis data[52]. The interannual variability of the June–October TC density (total days of TCs) of the JRA-25 reanalysis correlates well (at $r = 0.93$) with the RSMC Tokyo best track analysis[50], suggesting that the algorithm can assess the TC density reasonably well.

The monsoon trough (MT) index is defined as the area-integrated 850 hPa relative vorticity over a MT region (5°N–20°N, 130°E–180°)[40].

**Composite analysis of TC activity.** The composite TC activity analysis in Fig. 5d is based on the NINO3.4 and IOB indices. Composited events are chosen based on the observed NINO3.4 and IOB indices and are common to observational and prediction composites. High NINO3.4 SST years (1987, 1997 and 2015) are selected for the observed NINO3.4 index >1.5 std. dev. Positive IOB and low NINO3.4 years (1988, 1998 and 2010) are selected for the IOB index >0 and NINO3.4 index <−1 std. dev. In neutral to moderate ENSO cases, we classify years into two groups, namely, Groups A and B, by dividing years with the zero line of the bivariate linear regression equation for the TC density with the normalised NINO3.4 and IOB indices of the observations (Supplementary Information). The years of Group A are 1980, 1982, 1984, 1985, 1986, 1989, 1990, 1992, 1993, 1994, 1996, 2004, 2005, 2006, 2009 and 2012. The years of Group B are 1981, 1983, 1995, 2000, 2001, 2003, 2007, 2008, 2011, 2013, 2014 and 2016.

## Data availability

Data of the COBE-SST, JRA-55 reanalysis and RSMC Tokyo tropical cyclone best tracks used in this study are available from the Japan Meteorological Agency (http://ds.data.jma.go.jp/tcc/tcc/products/elnino/cobesst/cobe-sst.html, https://jra.kishou.go.jp/JRA-55/index.html, https://www.jma.go.jp/jma/jma-eng/jma-center/rsmc-hp-pub-eg/besttrack.html). The prediction data can be provided by the authors upon reasonable request.

## Code availability

The computer codes generated during the current study are available from the corresponding author on reasonable request.

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

## Acknowledgements

Y.T. is supported by JSPS KAKENHI Grant Numbers JP17K14395 and JP17K01223. Y.K. is supported by JSPS KAKENHI Grant Number JP18H01278, JP18H01281 and JP19H05703. Y.T., Y.K. and M.W. are supported by the Integrated Research Programme for Advancing Climate Models (TOUGOU) Grant Number JPMXD0717935561 and JPMXD0717935457 from the Ministry of Education, Culture, Sports, Science and Technology (MEXT), Japan. GPCP precipitation data were provided by the NOAA/OAR/ESRL PSD, Boulder, Colorado, USA, from their website (https://www.esrl.noaa.gov/psd/).

## Author contributions

Y.T. designed, performed and analysed the experiments. Y.T. draughted the initial manuscript. Y.K., M.W. and S.M. helped improve the analyses and writing. All authors discussed the results and contributed to the manuscript.

## Competing interests

The authors declare no competing interests.
