## [Peer Review File · Nature Communications]

REVIEWER COMMENTS

Reviewer #1 (Remarks to the Author):

This study obtained an interesting result: The IWNP monsoon can be predicted skillfully by a model ensemble prediction experiment with a lead time exceeding one year. This is an encouraging result, which is helpful for practice of monsoon prediction. The result may also trigger further research on air-sea interaction in the tropical Pacific and Indian Ocean. The authors explained the skillful long-lead-time prediction through the delayed ENSO influence mediated by Indian Ocean anomalies and large ensemble size. The paper has been well written. Therefore, I recommend accept after some revisions suggested as follows.

Specific comments:

1. The main predicting target of this study is WNP climate anomalies, including circulation and precipitation. Furthermore, the mechanisms of skillful prediction proposed by the authors are also mainly about the physical processes related to WNP climate anomalies. Therefore, I suggest more emphasis on the WNP climate in the revised paper, while less discussion on Asian monsoon (too broad concept). The Asian monsoon is a field with wide interest, but WNP monsoon is too. More and more studies demonstrate the importance of WNP monsoon, for instance, affecting East Asian monsoon and South Asian monsoon.
2. Although the rainfall over the Ganges, Amu Darya and Syr Darya basins seems to be predictable, it is hard to figure out the possible sources for the prediction skill, as mentioned by the authors. Without the possible mechanism, the prediction skill of rainfall over these regions may not be reliable. Furthermore, the Amu Darya and Syr Darya basins are not in the Asian summer monsoon regions. So I recommend removal of the results/discussion on these regions, focusing on the WNP as suggested above.
3. Regarding the mechanism, it is very interesting to see that the prediction skill decreases when summers following major El Nino events are excluded, which suggests the importance of major El Nino events in the skillful long-lead-time prediction of WNP climate anomalies. Because the Indian Ocean anomalies are suggested to play an important role in the delayed ENSO influence on WNP climate, it would be interesting to check whether the prediction skill of Indian Ocean anomalies (such as IOB SST) when summers following major El Nino events are excluded.
4. Prediction skill for temperatures in Figure 2 and Figure S4 should be given after detrending. Without detrending, the prediction skill would be highly artificial due to the global warming.
5. Figure S3 seems to be more important and valuable than Figure 3. The results shown in Figure 3 are only for two cases, while Figure S3 is about the general situation. So I suggest exchange the position of these figures between the text and supplementary information.

Reviewer #2 (Remarks to the Author):

Review: "Skillful predictions of the Asian summer monsoon one year ahead" by Takaya et al.

Using 52-member ensemble reforecasts with JMA/MRI-CPS2 for 37 years (1980-2016), the authors showed a skillful long-range prediction of the Asian summer monsoon and tropical cyclone activity, i.e., more than one year ahead. They argued that a successful simulation of the ENSO lifecycle and associated ocean-atmosphere interaction over the Indian Ocean and the northwestern Pacific in the second year is an essential factor for the skillful long-range prediction. Also, implications of the ensemble size for the further improvement of the prediction skill were also

discussed. This research work is significant for our understanding and predicting the Asian summer monsoon. While the overall quality of the paper is good, there are a few shortcomings that need to be addressed.

Some general and specific comments

1. L66-67: Please explain how the authors defined years of inactive WNP summer monsoon. With the naked eyes, the observed WNPM index in 2016 looks close to zero unlike 1998, whereas the predicted index in 2016 seems very close to that of 1998. It is not clear to me why 2016 is listed as an inactive WNP summer monsoon year, and it is hard for me to say that the model captured the inactive WNP summer monsoon in 2016.
2. Fig. 1: Correlation coefficient between ensemble mean forecast and observation increases after linear detrending in the equatorial eastern Pacific, but decreases over the tropical Indian Ocean (i.e., r versus r_d in Fig. 1b and c). Coupled seasonal forecast models in the North American Multimodel Ensemble (NMME) project, which were initialized from observed initial states, did not simulate the cooling tendency of the observed SST in the equatorial Pacific during the past two decades, but reasonably captured a warming trend over the Indian Ocean (e.g., Shin and Huang, 2019). Please show if this is also the case in this model, which may provide an additional explanation for a moderate prediction skill of the NINO3.4 index at a long-lead (i.e., 13-month lead), compared to IOB SST index. Also, it may help the authors to explain an enhanced (depressed) skill of NINO3.4 (IOB) index after linear detrending.
3. L76-78 and Fig. S2: Rebound of prediction skill of the IOB SST index in boreal fall (i.e., at 4~6 month leads in Fig. S2) may be attributed to the model's capability of Indian Ocean Dipole (IOD) prediction. This can be easily checked by plotting the prediction skill of IOD as Fig. S2. Also, it is noteworthy that the peak in the prediction skill of the IOB SST index in Fig. S2 appears in the spring of the second year when warming over the tropical Indian Ocean reaches the maximum during the decay year of El Nino.
4. Fig. 3 and Fig. 3S: Anomalies of each field in between Fig. 3 and Fig. 3S are opposite in terms of their sign, which may mislead readers. In particular, "anticyclone" in Fig. 3S (line 52 in the supplementary information) and "low surface pressure" in Fig. 3 (line 135 in the text) should be corrected. Perhaps, it would be better if they have the same sign of anomalies. Or briefly mention their opposite sign so that readers would not be confused.
5. L132-133 and L143-145: How well the model reproduced the two major El Ninos of 1997-1998 and 2015-2016 has not been shown. Time-longitude cross section (i.e., Hovmoller diagram) of SST and zonal wind stress anomalies averaged within 5S-5N in the Pacific Ocean or just monthly time series of SST anomalies averaged over NINO3.4 region for both observation and forecast with an envelope of all ensemble members would be helpful.
6. L106-148 & Fig. 3 or Fig. 3S: The results in Table 1 are most likely dependent upon the fact that the state-of-the art climate forecast systems may not be able to predict an onset of El Nino in their second year forecasts (e.g., 13-month lead) while they tend to reasonably well predict a development of El Nino in the first year and its demise and/or a transition from El Nino to La Nina in the second year. Thus, the results in Table 1 do not necessarily tell us whether the model can reproduce the IPOC development along the lifecycle of ENSO as described in the text. Besides, almost all spatial maps in this study including both Fig 3 and Fig. 3S show mean features only in boreal summer. Therefore, it would help to show how well the model captures the spatial and temporal evolution of air-sea interaction over the Indo-western Pacific basin associated with a decaying El Nino from its mature to demise phase (e.g., Shin et al. 2019; their Figs. 8-11), which may directly confirm the hypothesis in the text.
7. Grey shading in Figs. 1 and 5 represents a maximum-minim range of the ensemble predictions, but there seems no discussion about it. Since total 52 ensemble members have been used in this

study, I suggest the authors to use the ensemble spread, instead of the maximum-minimum range, which can more effectively measure an uncertainty of ensemble predictions, and add some discussion. Also, it may be interesting to see the relationship between the ensemble size and the ensemble spread as Fig. 4. Hopefully, this may provide a somewhat different perspective to the discussion of prediction skill dependency on ensemble size.

8. WNP TC predictions: What is the domain of the WNP for the analysis of TC predictions? Since many countries over the East Asia are used to experience Typhoons or their related heavy rainfall until early fall, I think TC is still active in September and even early October. As a consequence, analysis of WNP TC predictions for an extended time window (e.g., June-September) may be more appropriate.

Other minor comments

1. L39: In Fig. S1, please add a corresponding lead month in b and c, so readers can easily notice what "the 1st and 2nd summer" mean even before they would check the section of Methods.

2. The following references are apparently relevant.

L42-45: Shin et al. (2019)

L119-121: Wang et al. (2003) and Xiang et al. (2013)

3. L55-59 & L228-231, & Fig. 1: The x-axis of Fig. 1 (the 2nd summer) ranges from 1980 to 2016. It makes me assume that starting years of the prediction experiment are from 1979 to 2015, so the target years (i.e., the 2nd summer) cover 1980-2016. Is this right? Please clarify the period of the prediction experiment.

References:

Shin et al., 2019: Improved seasonal predictive skill and enhanced predictability of the Asian summer monsoon rainfall following ENSO events in NCEP CFSv2 hindcasts. *Climate Dyn.*, 52, 3079-2098, doi:10.1007/s00382-018-4316-y

Shin and Huang, 2019: A spurious warming trend in the NMME equatorial Pacific SST hindcasts. *Climate Dyn.*, 53, 7287-7303, <https://doi.org/10.1007/S00382-017-3777-8>.

Xiang et al., 2013: How can anomalous western North Pacific subtropical high intensify in late summer? *Geophys. Res. Lett.* 40:2349-2354

Wang et al., 2003: Atmosphere-warm ocean interaction and its impacts on Asian-Australian monsoon variation. *J. Clim.* 16:1195-1211

Response to reviewers' comments

We wish to express our appreciation to the reviewers for all the constructive comments that helped us to improve the manuscript. Two manuscript files are attached, one with track changes and the other without the track changes. In the point-by-point response, the reviewer's comments are written in black, followed by our response in blue. The line numbers in our reply refer to those of the PDF file without track changes.

Response to Reviewer #1:

This study obtained an interesting result: The IWNP monsoon can be predicted skillfully by a model ensemble prediction experiment with a lead time exceeding one year. This is an encouraging result, which is helpful for practice of monsoon prediction. The result may also trigger further research on air-sea interaction in the tropical Pacific and Indian Ocean. The authors explained the skillful long-lead-time prediction through the delayed ENSO influence mediated by Indian Ocean anomalies and large ensemble size. The paper has been well written. Therefore, I recommend accept after some revisions suggested as follows.

We would like to express our appreciation to the reviewer for all the constructive comments that helped us to improve the manuscript. In accordance with the reviewer's suggestions, we have thoroughly revised the manuscript. We believe that additional analyses of the detrended prediction skill and predictability, and the lag composite help to further convince readers of the successful prediction and supporting mechanisms.

Specific comments:

1. The main predicting target of this study is WNP climate anomalies, including circulation and precipitation. Furthermore, the mechanisms of skillful prediction proposed by the authors are also mainly about the physical processes related to WNP climate anomalies. Therefore, I suggest more emphasis on the WNP climate in the revised paper, while less discussion on Asian monsoon (too broad concept). The Asian monsoon is a field with wide interest, but WNP monsoon is too. More and more studies demonstrate the importance of WNP monsoon, for instance, affecting East Asian monsoon and South Asian monsoon.

In the revised manuscript, more focus has been put on the WNP monsoon and IPOC mode and their representation in the model. To this aim, we have added lag-composite analysis to support

the model's representation of the IPOC evolution following major El Niño events. (Fig. S7). We have also discussed this result in Lines 153–156 in the main text and Lines 160–164 in the Supplementary Information. Following the reviewer's comment #2, we have removed the discussion on the Amu Darya and Syr Darya basin precipitation and removed drawing of these river basins from Fig. 2.

Lines 153–156: The lag composites of the observations and predictions after the major El Niños further support that the model can reproduce and predict the IPOC development along the lifecycle of ENSO (Supplementary Information, Fig. S7).

Supplementary Information, Lines 160–164: The model's ability to reproduce the IPOC development along the lifecycle of ENSO is evaluated by the lag composite analysis after the major El Niños (Fig. S7). The mechanisms of the IPOC evolution after El Niño (see details in the main text) are reasonably reproduced in the model, including the basin-wide warming of IO after El Niño and associated atmospheric conditions^{19,23,27}.

2. Although the rainfall over the Ganges, Amu Darya and Syr Darya basins seems to be predictable, it is hard to figure out the possible sources for the prediction skill, as mentioned by the authors. Without the possible mechanism, the prediction skill of rainfall over these regions may not be reliable. Furthermore, the Amu Darya and Syr Darya basins are not in the Asian summer monsoon regions. So I recommend removal of the results/discussion on these regions, focusing on the WNP as suggested above.

Considering the reviewer's suggestion, we have removed the sentence of the prediction of precipitation over the Amu Darya and Syr Darya basins. Regarding the Ganges region, a previous study (Chowdary et al. 2019) suggested the rainfall in the Ganges region is associated with the IPOC mode. We have mentioned this in Lines 95–96 and retained the Ganges rainfall results.

Lines 95–96: Previous studies suggested that precipitation in the Ganges river basin is associated with Asian monsoon variability^{32,33}.

3. Regarding the mechanism, it is very interesting to see that the prediction skill decreases when summers following major El Niño events are excluded, which suggests the importance of major El Niño events in the skillful long-lead-time prediction of WNP climate anomalies. Because the Indian Ocean anomalies are suggested to play an important role in the delayed ENSO influence on WNP climate, it would be interesting to check whether the prediction skill of Indian Ocean

anomalies (such as IOB SST) when summers following major El Niño events are excluded.

We have presented the prediction skill results of IOB SST in Table S1. The result indicates that the IOB SST skill decreases when summers following El Niño events are excluded. We have mentioned this point in the main text (Lines 132–134).

Lines 132–134: The above hypothesis is confirmed by a simple skill evaluation: the prediction performance of IOB SST and associated WNP summer monsoon tends to decrease when summers following major El Niños are excluded (Supplementary Information, Table S1).

4. Prediction skill for temperatures in Figure 2 and Figure S4 should be given after detrending. Without detrending, the prediction skill would be highly artificial due to the global warming.

We have added the prediction skill after detrending (Fig. S5) and the potential predictability after detrending (Fig. S4b). As seen in Fig. S5, the model has meaningful prediction skill after detrending in some regions, suggesting that the model has capability to predict the interannual variability, not only the trend. The results indicate that the prediction skill of surface temperature and SST is increased due to the trend signal, consistent with previous studies (e.g., Doblas-Reyes et al. 2006), whereas that of precipitation and 850-hPa zonal wind is virtually unaffected (Figs. 2 and S5). Furthermore, the potential predictability computed after detrending (Fig. S4b) indicates relatively large influence of the trend on temperature predictability over land regions (e.g., Eurasia) and the Indian Ocean compared with precipitation predictability, consistent with the actual prediction skill (Fig. S5). We have discussed these points in Line 103–107 in the main text, and Lines 127–132 and 135–142 in Supplementary Information.

Lines 103–107: Notably, the model retains meaningful prediction skill after linear detrending, indicating its capability to predict the interannual variability (Supplementary Information, Fig. S5). Moreover, the contribution of the trend to the actual prediction skill is also consistent with its contribution to the potential predictability (Fig. 2; Supplementary Information, Figs. S4 and S5).

Supplementary Information, Lines 127–132: We also investigated the potential predictability after detrending to assess the skill gains from the long-term trend. It is evident that the potential predictability of surface temperature decreases by detrending, in particular over Eurasia, consistent with the previous studies⁵⁷. By contrast, the detrending much less affects

the potential predictability for 850-hPa zonal wind and precipitation. These characteristics agree with the actual prediction skill of the model (Figs. 2 and S5).

Supplementary Information, Lines 135–142: The prediction skill (correlation scores) is increased due to the long-term trend observed in the verification period⁵⁵. Better representing the trend in the model is important to establish the skilful seasonal prediction, while it is worth evaluating the prediction skill after detrending to confirm whether the model has capability to predict the interannual variability in addition to the trend. Figure S5 presents the second-summer prediction skill after detrending. The results (Figs. 2 and S5) indicate that the prediction skill of surface temperature is aided by the trend signal consistent with previous studies⁵⁵, whereas that of precipitation and 850-hPa zonal wind is virtually unaffected.

Figure S5 | As Fig. 2, but the prediction skill after linear detrending.

Figure S4 | Potential predictability (correlation coefficients) of the second summer predictions (a) without and (b) with detrending. Correlation coefficients between the surrogate observations and ensemble mean predictions for the (top) 850-hPa zonal wind, (middle) 2-m air temperature over land and SST over the ocean, and (bottom) precipitation. Stippled regions are statistically significant at the 5% level according to a bootstrap method (10,000 resamplings).

5. Figure S3 seems to be more important and valuable than Figure 3. The results shown in Figure 3 are only for two cases, while Figure S3 is about the general situation. So I suggest exchange the position of these figures between the text and supplementary information.

Considering the reviewers' comments, we have replaced the lag composite figures for the two years with those for all the major decaying El Niño summers (1983, 1992, 1998, 2003, 2010 and 2016; Fig. 3) to present common features after the major El Niños. Here, the major El Niños were selected as NDJ NINO3.4 SST anomaly > 1 std. dev. As seen in Fig. 3 and Fig. S3, these two figures display similar patterns and support the same conclusion, thus, we have presented the composite figures (Fig.3) in the main text, and include Fig. S3 in the Supplementary Information. We have revised Lines 139–142, accordingly.

Lines 139–142: Composite maps for boreal summers after major El Niños (observed NINO3.4 index > 1 std. dev. in preceding November–January; 1983, 1992, 1998, 2003, 2010 and 2016; Fig. 3) further substantiate that the ENSO–IPOC coupling conveys successful

second-summer predictions for the Indo-WNP and Asian climate.

Figure 3 | The IPOC mode and its second summer prediction. Composite anomalies of the (a,b) observations and (c,d) JMA/MRI-CPS2 13-month lead prediction for summers following major El Niño events (1983, 1992, 1998, 2003, 2010 and 2016; see text for definition). (a,c) Precipitation (colours) and sea level pressure (contours with an interval of 0.3 hPa; red for positive, grey for zero, and blue for negative). (b,d) SST over the ocean and 2-m air temperature over land (colours) and 500-hPa geopotential height (contours with an interval of 3 m; red for positive, grey for zero, and green for negative). Stippled regions are statistically significant at the 5% level based on a bootstrap method (10,000 resamplings).

Figure S3 | Climate anomalies associated with WNP summer monsoon variability. Summer (JJA) average anomalies of (a,c,e) precipitation, sea level pressure (SLP; contours with an interval of 0.2 hPa; red for positive, grey for zero, and blue for negative) and 850-hPa wind. (b,d,f) SST and 2-m air temperature over land (LST) and 500-hPa height (contours with an interval of 2 m; red for positive, grey for zero, and green for negative) are regressed on the standardized WNP index with its sign being flipped. Results for the (a,b) observations, (c,d) first summer prediction and (e,f) second summer prediction.

Reference:

Chowdary, J. S., Patekar, D., Srinivas, G., Gnanaseelan, C. & Parekh, A. Impact of the Indo-Western Pacific Ocean Capacitor mode on South Asian summer monsoon rainfall. *Clim. Dyn.* **53**, 2327–2338 (2019).

Doblas-Reyes, F. J., R. Hagedorn, T. N. Palmer, J.-J. Morcrette. Impact of increasing greenhouse gas concentrations in seasonal ensemble forecasts. *Geophys. Res. Lett.* **33**, L07708 (2006).

Response to Reviewer #2:

Review: “Skilful predictions of the Asian summer monsoon one year ahead”; by Takaya et al.

Using 52-member ensemble reforecasts with JMA/MRI-CPS2 for 37 years (1980-2016), the authors showed a skillful long-range prediction of the Asian summer monsoon and tropical cyclone activity, i.e., more than one year ahead. They argued that a successful simulation of the ENSO lifecycle and associated ocean-atmosphere interaction over the Indian Ocean and the northwestern Pacific in the second year is an essential factor for the skillful long-range prediction. Also, implications of the ensemble size for the further improvement of the prediction skill were also discussed. This research work is significant for our understanding and predicting the Asian summer monsoon. While the overall quality of the paper is good, there are a few shortcomings that need to be addressed.

We would like to express our appreciation to the reviewer for the constructive comments that helped us to improve the manuscript. In accordance with the reviewer’s comments and suggestions, we have thoroughly and carefully revised the manuscript. We have revised the composite figures, and added the lag composite analysis (a sequence of the IPOC development after El Niños) and IOD skill to support our conclusions.

Some general and specific comments

1. L66-67: Please explain how the authors defined years of inactive WNP summer monsoon. With the naked eyes, the observed WNPM index in 2016 looks close to zero unlike 1998, whereas the predicted index in 2016 seems very close to that of 1998. It is not clear to me why 2016 is listed as an inactive WNP summer monsoon year, and it is hard for me to say that the model captured the inactive WNP summer monsoon in 2016.

Considering the reviewers’ comments, we have removed this sentence. In addition, we have replaced the composites of 1998 and 2016 with those for all the decaying El Niño summers (1983, 1992, 1998, 2003, 2010 and 2016; Fig. 3) to discuss common features in summers after major El Niños. The revised composite maps for more decaying El Niño summers provide statistically more robust and typical features. The results indicate that the inactive WNP monsoon in decaying El Niño summers is reasonably captured by the model. We have revised Lines 139–142, accordingly.

Lines 139–142: Composite maps for boreal summers after major El Niños (observed

NINO3.4 index > 1 std. dev. in preceding November–January; 1983, 1992, 1998, 2003, 2010 and 2016; Fig. 3) further substantiate that the ENSO–IPOC coupling conveys successful second-summer predictions for the Indo-WNP and Asian climate.

Figure S3 | Climate anomalies associated with WNP summer monsoon variability. Summer (JJA) average anomalies of (a,c,e) precipitation, sea level pressure (SLP; contours with an interval of 0.2 hPa; red for positive, grey for zero, and blue for negative) and 850-hPa wind. (b,d,f) SST and 2-m air temperature over land (LST) and 500-hPa height (contours with an interval of 2 m; red for positive, grey for zero, and green for negative) are regressed on the standardized WNP index with its sign being flipped. Results for the (a,b) observations, (c,d) first summer prediction and (e,f) second summer prediction.

2. Fig. 1: Correlation coefficient between ensemble mean forecast and observation increases after linear detrending in the equatorial eastern Pacific, but decreases over the tropical Indian Ocean (i.e., r versus r_d in Fig. 1b and c). Coupled seasonal forecast models in the North American Multimodel Ensemble (NMME) project, which were initialized from observed initial states, did not simulate the cooling tendency of the observed SST in the equatorial Pacific during the past two decades, but reasonably captured a warming trend over the Indian Ocean (e.g., Shin and Huang, 2019). Please show if this is also the case in this model, which may provide an additional

explanation for a moderate prediction skill of the NINO3.4 index at a long-lead (i.e., 13-month lead), compared to IOB SST index. Also, it may help the authors to explain an enhanced (depressed) skill of NINO3.4 (IOB) index after linear detrending.

As seen in Fig.1, the model has a spurious warming trend in the central-eastern equatorial Pacific in the second summer. By contrast, the model exhibits a stronger warming SST trend over the tropical Indian Ocean than the observations. Thus, this contrast reasonably explains the prediction skill difference of IOB and NINO3.4 after detrending, as the reviewer pointed out. We have discussed this point in Lines 78–81. For your reference, the spatial distribution of the trend of SST is shown in Fig. A (see also Fig. 2 of Shin and Huang (2019)).

Lines 78–81: It is also noted that the model presents higher prediction skill of NINO3.4 SST with linear detrending than without, due to an overestimate of the warming trend of NINO3.4 SST in the model (Fig. 1b), consistent with a multi-model ensemble coupled seasonal predictions³¹.

Figure A | Observed and predicted trends for SST in summer (JJA) during 1980-2016. Trends of (a) observation and predictions for the (b) first (lead time of 1 month) and (c) second summer (lead time of 13 months).

3. L76-78 and Fig. S2: Rebound of prediction skill of the IOB SST index in boreal fall (i.e., at 4~6 month leads in Fig. S2) may be attributed to the model's capability of Indian Ocean Dipole (IOD) prediction. This can be easily checked by plotting the prediction skill of IOD as Fig. S2. Also, it is noteworthy that the peak in the prediction skill of the IOB SST index in Fig. S2 appears in the spring of the second year when warming over the tropical Indian Ocean reaches the maximum during the decay year of El Niño.

Thank you for your insightful comment. We have added the IOD prediction skill (Dipole Mode Index; DMI) as a function of lead time (Fig. S2). We found moderate correlations ($r \sim 0.5-0.6$) of DMI (October–December) with IOBW (October–December, and subsequent seasons) in the observation and prediction (Fig. B). Thus, as the reviewer suggested, the seasonality of the

prediction skill of DMI is likely to be partly associated with that of IOB SST. Moreover, IOD is one of key processes, which contributes to generate the IOB warming (Xie et al. 2016). We have noted this in Lines 32–38 in the Supplementary Information. We have also mentioned that the prediction skill of the IOB SST index (Fig. S2) peaks in boreal spring, when salient warming of IOB SST appears in decaying El Niño years (Supplementary Information, Lines 28–29).

Supplementary Information, Lines 28–29: The prediction skill of the IOB SST peaks in spring, when salient warming of IOB SST occurs in decaying El Niño years^{23,27}.

Supplementary Information, Lines 32–38: The Dipole Mode Index (DMI), which is defined as the difference between SST anomalies averaged over key western (10°N–10°S, 50°E–70°E) and eastern (Eq. –10°S, 90°E–110°E) regions and represents the Indian Ocean Dipole (IOD), was evaluated⁵⁴. The prediction skill of DMI exhibits the characteristic seasonal dependency with the first peak in boreal autumn, when the IOD matures, and the second peak in boreal spring. The rebound of the IOB SST prediction skill is considered to reflect the delayed influence of ENSO and IOD²⁷.

Figure S2 | Prediction skills of NINO3.4 SST, IOB SST and DMI as a function of lead time. Correlation coefficient between observations and predictions. Curves indicate correlations with ensemble mean predictions. The first and second summer (June-July-August; JJA) prediction skills are denoted with closed and open circles, respectively. The uncertainty ranges are based on a bootstrap method (10,000 resamplings).

Figure B | Lead-lag correlations of 3-month average DMI (OND) with 3-month averages of IOBW. Linear detrending was applied to both indices before computing correlations. Red and blue lines indicate correlations for the observation and prediction, respectively.

4. Fig. 3 and Fig. 3S: Anomalies of each field in between Fig. 3 and Fig. 3S are opposite in terms of their sign, which may mislead readers. In particular, “anticyclone” in Fig. 3S (line 52 in the supplementary information) and “low surface pressure” in Fig. 3 (line 135 in the text) should be corrected. Perhaps, it would be better if they have the same sign of anomalies. Or briefly mention their opposite sign so that readers would not be confused.

Following the reviewer’s suggestion, we have flipped the sign in Fig. S3 and have corrected the figure caption and text accordingly (Supplementary Information, Lines 58–62).

Supplementary Information, Lines 58–62: These maps correspond to the conditions typical of El Niño following summers when the WNP summer monsoon tends to be anomalously weak. The observations feature an anomalous lower-tropospheric anticyclone and decreased rainfall in the tropical WNP, increased rainfall over the Indian Ocean and Maritime Continent, and the Pacific–Japan teleconnection pattern³⁶.

Figure 3 | The IPOC mode and its second summer prediction. Composite anomalies of the (a,b) observations and (c,d) JMA/MRI-CPS2 13-month lead prediction for summers following major El Niño events (1983, 1992, 1998, 2003, 2010 and 2016; see text for definition). (a,c) Precipitation (colours) and sea level pressure (contours with an interval of 0.3 hPa; red for positive, grey for zero, and blue for negative). (b,d) SST over the ocean and 2-m air temperature over land (colours) and 500-hPa geopotential height (contours with an interval of 3 m; red for positive, grey for zero, and green for negative). Stippled regions are statistically significant at the 5% level based on a bootstrap method (10,000 resamplings).

5. L132-133 and L143-145: How well the model reproduced the two major El Niños of 1997-1998 and 2015-2016 has not been shown. Time-longitude cross section (i.e., Hovmoller diagram) of SST and zonal wind stress anomalies averaged within 5S-5N in the Pacific Ocean or just monthly time series of SST anomalies averaged over NINO3.4 region for both observation and forecast with an envelope of all ensemble members would be helpful.

Considering the reviewer's comment #1 and #5, we have presented the composite maps of all the major El Niño-decay summers, not the two summers (Fig. 3). The major El Niños were defined objectively as events with observed NDJ-mean NINO3.4 SST index exceeding 1 std. dev. (Please note that the back-to-back El Niño case of 1986–1988 was excluded as Shin et al. 2019.) The prediction skill of NINO3.4 as a function of lead time is presented in Fig.S2. For your reference, NINO3.4 SST predictions for the major El Niño events are presented in Fig. C. As seen in Fig. C, the two major El Niños of 1997–1998 and 2015–2016 were predicted well.

Figure C | Observed (red) and predicted (blue) NINO3.4 SST anomalies (K) for the major El Niño events. Three-month average NINO3.4 SST anomalies are presented and the x-axis indicates lead time. The calendar month of the x-axis indicate the first month of the three-month average periods. Shades indicate the maximum-minimum ranges.

6. L106-148 & Fig. 3 or Fig. 3S: The results in Table 1 are most likely dependent upon the fact that the state-of-the-art climate forecast systems may not be able to predict an onset of El Niño in their second year forecasts (e.g., 13-month lead) while they tend to reasonably well predict a development of El Niño in the first year and its demise and/or a transition from El Niño to La Niña in the second year. Thus, the results in Table 1 do not necessarily tell us whether the model can reproduce the IPOC development along the lifecycle of ENSO as described in the text. Besides, almost all spatial maps in this study including both Fig 3 and Fig. 3S show mean features only in boreal summer. Therefore, it would help to show how well the model captures the spatial and temporal evolution of air-sea interaction over the Indo-western Pacific basin associated with a decaying El Niño from its mature to demise phase (e.g., Shin et al. 2019; their Figs. 8-11), which may directly confirm the hypothesis in the text.

To support our conclusion, we have added the lag composite analysis of consecutive seasons (Fig. S7). The lag composites indicate that the model can reproduce the IPOC development (Indian Ocean warming and associated atmospheric circulation) along the lifecycle of ENSO. We have also checked the results of lag regression analysis between the NINO3.4 SST index of November–January and conditions in following summers, and found that the results are consistent with the lag composite analysis. These results suggest that the model can reproduce and predict the IO warming after El Niños and IPOC development along the lifecycle of ENSO. We have mentioned this in Lines 153–156 in the main text and Lines 160–164 in the Supplementary Information.

Lines 153–156: The lag composites of the observations and predictions after the major El Niños further support the model capability of reproducing and predicting the IPOC development along the lifecycle of ENSO (Supplementary Information, Fig. S7).

Supplementary Information, Lines 160–164: The model’s ability to reproduce the IPOC development along the lifecycle of ENSO is evaluated by the lag composite analysis after the major El Niños (Fig. S7). The mechanisms of the IPOC evolution after El Niño (see details in the main text) are reasonably reproduced in the model, including the basin-wide warming of IO after El Niño and associated atmospheric conditions^{19,23,27}.

Figure S7 | A sequence of IPOC evolution after major El Niño events. Composite anomalies of the (a,c,e) observations and (b,d,f) JJA/MRI-CPS2 prediction for (a,b) winters, (c,d) springs and (e,f) summers following the major El Niño events. Precipitation (colours), sea surface temperature (contours) and 850-hPa wind (vectors). All the fields are drawn after detrending. Contours are drawn for ± 0.2 , ± 0.4 , ± 0.8 , ± 1.6 , ± 2.4 . Stippled regions are statistically significant at the 5% level based on a bootstrap method (10,000 resamplings). Vectors are plotted where zonal or meridional wind are statistically significant at the 5% level based on a bootstrap method (10,000 resamplings).

7. Grey shading in Figs. 1 and 5 represents a maximum-minimum range of the ensemble predictions, but there seems no discussion about it. Since total 52 ensemble members have been used in this study, I suggest the authors to use the ensemble spread, instead of the maximum-minimum range, which can more effectively measure an uncertainty of ensemble predictions, and add some discussion. Also, it may be interesting to see the relationship between the ensemble size and the ensemble spread as Fig. 4. Hopefully, this may provide a somewhat different perspective to the discussion of prediction skill dependency on ensemble size.

We have noted in the main text that the maximum-minimum range of the ensemble prediction reasonably cover the observations by virtue of the large ensemble size (52 members) in Lines 63–64 and 193–194. Maximum-minimum ranges of the ensemble prediction represent a sort of the uncertainty range of the ensemble prediction and they are often used to indicate the range of the ensemble prediction. Considering the reviewer’s comment, we have also added the interquartile ranges (25–75%), which is also often used for expressing the uncertainty range, in Figs. 1, 5 and S6.

Regarding the spread-skill relationship, we have examined it for the WNP monsoon index, and have found no meaningful correlation of the absolute error (after bias correction) with the spread (and interquartile range). This may be primarily due to the relatively low skill (typically $r \sim 0.5$) and the fact that origins of seasonal predictions are different from those of short-range predictions. Although, the spread is considered as an indicator of the expected skill in the short-range ensemble prediction, several prior studies have found that it is not the case for the low-skill predictions beyond about 10 days (Barker 1991, Whitaker and Lough 1998). For this reason, the spread is considered to be useless from a practical standpoint in the seasonal prediction (e.g., Doblas-Reyes et al. 2000, Tang et al. 2008). Exploring the dependency of the spread-skill relationship on the ensemble size and better ways to assess the flow-dependent predictability in the seasonal prediction is an important research topic (e.g., Grit and Mass 2007, Tang et al. 2008), but this topic is beyond the scope of this study, and needs future work.

Lines 63–64: Ensemble envelopes (maximum–minimum ranges) generally encompass the observation by virtue of the large ensemble size.

Lines 193–194: The ensemble envelopes generally encompass the observed TC activity and MT index.

8. WNP TC predictions: What is the domain of the WNP for the analysis of TC predictions? Since many countries over the East Asia are used to experience Typhoons or their related heavy rainfall until early fall, I think TC is still active in September and even early October. As a consequence, analysis of WNP TC predictions for an extended time window (e.g., June-September) may be more appropriate.

Thank you for your comment. We have added the domain of the TC prediction (Line 180). Regarding the time window of the WNP TC prediction, after careful consideration of the reviewer's comment, we have decided to present the results for the current time window (June–August). We agree with the reviewer's comment that the full WNP TC active season is June–September or June–October (please see Fig.1a of Du et al. 2011 for the TC number). Thus, we have noted that we analyzed the first half of the WNP TC season in this study (Lines 180–181). There are mainly three reasons for the choice of the TC time window (June–August) in this study.

- (1) The prediction skill of TC activity is lower for July–September ($r = 0.35$) in the second year than June–August ($r = 0.39$) due to longer lead time.
- (2) According to Fig. 1A in Kosaka et al. (2013), the influence of the preceding ENSO on the TC genesis peaks in June and weakens through subsequent months (July, August, September). In particular, the correlation between NINO3.4 SST in preceding winter (November–January) and the TC genesis in September is weak and not statistically significant. This indicates that the influence of the preceding ENSO appears more clearly in June–August than July–September.
- (3) Consistently, the previous study (Takaya et al. 2017) found that the seasonal prediction of the WNP TC genesis is more skilful for early summer, than for late summer, due to the stronger effect of the IO or preceding ENSO.

Considering these reasons, we have retained the original TC time window. Instead, we have mentioned the skill for July–September and added a brief discussion in Lines 180–185. It is noted that TCs in June–August affect Southeast and East Asian countries and IO and ENSO modulate the TC density (Fig. 5), thus our results are still valuable even though the period does not cover the full TC season.

Lines 179–185: The model can make predictions of the WNP TC density averaged over the WNP (0° – 60° N, 100° W– 180°) during June–August, which corresponds to the first half of the WNP TC season³⁴, with a highly significant skill ($r = 0.67$, $p < 0.001$) for the first summer (Fig. 5a) and a moderate skill ($r = 0.39$, $p = 0.017$) for the second summer (Fig. 5b). We note that the skill is slightly lower for July–September in the second year ($r = 0.35$, $p = 0.035$) than for June–August, due to gradual dissipation of delayed ENSO influence and skill decline

with a lead time^{23,35}.

Other minor comments

1. L39: In Fig. S1, please add a corresponding lead month in b and c, so readers can easily notice what “the 1st and 2nd summer” mean even before they would check the section of Methods.

As suggested, we have added labels of the lead month in Fig. S1b and c.

2. The following references are apparently relevant.

L42-45: Shin et al. (2019)

L119-121: Wang et al. (2003) and Xiang et al. (2013)

We have added the suggested references (Lines 44–45, 126).

2. L55-59 & L228-231, & Fig. 1: The x-axis of Fig. 1 (the 2nd summer) ranges from 1980 to 2016. It makes me assume that starting years of the prediction experiment are from 1979 to 2015, so the target years (i.e., the 2nd summer) cover 1980-2016. Is this right? Please clarify the period of the prediction experiment.

Yes it is. The same target period (1980–2016) was evaluated in the verification of the first and second summers. In other words, starting years of the predictions are from 1980 to 2016 (1979 to 2015) for the first (second) summer predictions. We have mentioned this point clearly (Lines 55–59).

Lines 55–59: A set of 52-member ensemble hindcasts starting from every April was established for the summer seasons for 37 years from 1980 to 2016. We particularly focused on the prediction skill for the second-year boreal summer (June–August with a 13-month lead) against historical observations and reanalysis (Methods).

References:

- Barker, T. W. The Relationship between Spread and Forecast Error in Extended-range Forecasts. *J. Clim.* **4**, 733–742 (1991).
- Doblas-Reyes, F.J., Déqué, M. & Piedelievre, J.-P. Multi-model spread and probabilistic seasonal forecasts in PROVOST. *Q.J.R. Meteorol. Soc.* **126**, 2069–2087 (2000).
- Du, Y., Yang, L. & Xie, S.-P. Tropical Indian Ocean influence on Northwest Pacific tropical cyclones in summer following strong El Niño. *J. Clim.* **24**, 315–322 (2010).

- Grimit, E. P. & Mass. C. F. Measuring the Ensemble Spread–Error Relationship with a Probabilistic Approach: Stochastic Ensemble Results. *Mon. Wea. Rev.* **135**, 203–221 (2007).
- Shin, C.-S., Huang, B., Zhu, J., Marx, L. & Kinter III. J. L. Improved seasonal predictive skill and enhanced predictability of the Asian summer monsoon rainfall following ENSO events in NCEP CFSv2 hindcasts. *Clim. Dyn.* **52**, 3079–3098 (2019).
- Shin, C.-S. & Huang, B. A spurious warming trend in the NMME equatorial Pacific SST hindcasts. *Clim. Dyn.* **53**, 7287–7303 (2019).
- Tang, Y., Kleeman, R. & Moore. A. M. Comparison of Information-Based Measures of Forecast Uncertainty in Ensemble ENSO Prediction. *J. Clim.* **21**, 230–247 (2008).
- Wang, B., Wu, R. & Li, T. Atmosphere–warm ocean interaction and its impacts on Asian–Australian monsoon variation. *J. Clim.* **16**, 1195–2013 (2003).
- Whitaker, J. S. & Lough, A. F. The Relationship between Ensemble Spread and Ensemble Mean Skill. *Mon. Wea. Rev.* **126**, 3292–3302 (1998).
- Xiang, B., Wang, B., Yu, W. & Xu, S. How can anomalous western North Pacific subtropical high intensify in late summer? *Geophys. Res. Lett.* **40**, 2349–2354 (2013).

REVIEWERS' COMMENTS

Reviewer #1 (Remarks to the Author):

The authors have revised the manuscript according to my comments. I have no further comments for improvement, and recommend accept.

Reviewer #2 (Remarks to the Author):

I would like to thank the authors for their responses and additions to the manuscript. In the revised manuscript, all my comments, suggestions and questions have been addressed in detail, including some new diagnostics (e.g., lag composite analysis for the IPOC development after El Nino and evaluation of IOD prediction skill).

Here are just a couple of minor comments I have come across while reading through the revised manuscript.

1. Fig. S7 and Fig. 3: I agree that overall, the model reasonably reproduced the IPOC development along the ENSO lifecycle. However, one noticeable disagreement between observation and prediction is long-lasting warm SST anomalies over the tropical western Pacific until the subsequent summer in the model, which may account for enhanced rainfall anomalies there. Unlike southwest-northeast contrast of precipitation and SLP anomalies in the observation, therefore, it seems to make more zonal pattern of low pressure (enhanced rainfall) anomalies along the Equator in the model, sandwiched by high pressure (reduced rainfall) anomalies in the subtropic (or off-Equator) regions of both Hemispheres.

2. There are no latitude and longitude for all spatial maps.

Response to reviewers' comments

We wish to express our appreciation to the reviewers for all the constructive comments that helped us to improve the manuscript. Two manuscript files are attached, one with track changes and the other without the track changes. In the point-by-point response, the reviewer's comments are written in black, followed by our response in blue. The line numbers in our reply refer to those of the PDF file without track changes.

Response to Reviewer #1:

Reviewer #1 (Remarks to the Author):

The authors have revised the manuscript according to my comments. I have no further comments for improvement, and recommend accept.

We would like to express our appreciation to the reviewer for all the constructive comments and suggestions that helped us to improve the manuscript.

Response to Reviewer #2:

I would like to thank the authors for their responses and additions to the manuscript. In the revised manuscript, all my comments, suggestions and questions have been addressed in detail, including some new diagnostics (e.g., lag composite analysis for the IPOC development after El Nino and evaluation of IOD prediction skill).

Here are just a couple of minor comments I have come across while reading through the revised manuscript.

We would like to express our appreciation to the reviewer for all the constructive comments that helped us to improve the manuscript. In accordance with the reviewer's comments and suggestions, we have revised Figures 2, 3, 5, S3, S4, S5 and S7, and added a sentence that was suggested.

1. Fig. S7 and Fig. 3: I agree that overall, the model reasonably reproduced the IPOC development

along the ENSO lifecycle. However, one noticeable disagreement between observation and prediction is long-lasting warm SST anomalies over the tropical western Pacific until the subsequent summer in the model, which may account for enhanced rainfall anomalies there. Unlike southwest-northeast contrast of precipitation and SLP anomalies in the observation, therefore, it seems to make more zonal pattern of low pressure (enhanced rainfall) anomalies along the Equator in the model, sandwiched by high pressure (reduced rainfall) anomalies in the subtropic (or off-Equator) regions of both Hemispheres.

We appreciate the reviewer's comment on this point. As the reviewer pointed out, we see the transition from El Niños to La Niñas seems to be somewhat weaker than that observed and the warm SST anomaly remains in the El Niño following summers (Figures 3d and S7f). Due to this difference of the SST anomaly, the model presents more zonal anomaly pattern of precipitation and SLP in the equatorial western Pacific (Figures 3c and 7f). We touched on this point in Lines 155–157.

Lines 155–157: The model, however, presents a tendency of a weaker transition from El Niño to La Niña than that observed, which may limit the prediction skill of the WNP summer monsoon (Fig. 3).

2. There are no latitude and longitude for all spatial maps.

In accordance with the reviewer's comment, we have added latitude and longitude labels in Figures 2, 3, 5, S3, S4, S5 and S7.